# LEARNING IDENTIFIABLE CONCEPTS FOR COMPOSITIONAL IMAGE GENERATION

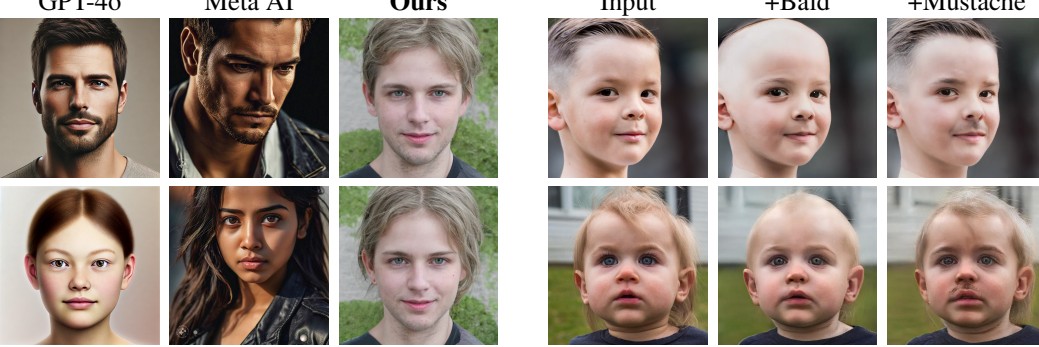

(a) Generation results with *male face with 5 o'clock shadow* and *young girl face with 5 o'clock shadow*.

(b) Real image editing results by our method. We are able to generate some uncommon concept compositions, such as little boy with mustache.

Figure 1: Comparison of generation and editing results. The strongest AI foundational models fails to composite the *5 o'clock shadow* concept with the *young girl* concept. By contrast, our model learns to add the shadow to male and girl face in a compositional way.

## ABSTRACT

Humans have the ability to decompose objects into parts and relationships and create new objects by properly combining existing concepts. However, enabling machines to achieve this in real-world tasks remains a challenge. In this paper, we investigate how to teach machines compositional image generation through learning identifiable concepts. To derive concepts from attribute labels, we formulate the minimal change principle and propose a method to limit the information introduced by each label. Additionally, to address dependent attribute labels (with causal influences in between or common causes behind them), we present a causal conditioning approach to disentangle concepts from these correlations. Our framework enhances data efficiency, interpretability, and control, while enabling sampling from unseen combinations. We validate our method on various compositional image generation and editing tasks, demonstrating its effectiveness through superior performance.

## 1 INTRODUCTION

Despite the recent significant progress in artificial intelligence, machine learning systems still have not yet been able to fully capture human conceptual learning. Humans are able to parse objects into parts and relationships and create new objects with existing concepts (Lake et al., 2015). However, it can be challenging for the machines to achieve that in compositional image generation task, even for the most powerful AI models, e.g., GPT-4o and Meta AI. As shown in Figure 1, they fail to add *5 o'clock shadow* to *young girl face* while they are able to generate *male face* with *5 o'clock shadow*. However, human can easily imagine a young girl face with the 5 o'clock shadow. The failures show that machine learning systems still have difficulty processing human face into different concepts and cannot composite the *young girl* concept with *5 o'clock shadow* concept together like human does.

There have been many efforts on compositing different objects for image generation, such as adding objects to scene images (Tan et al., 2019; Lin et al., 2018; Tan et al., 2019; Song et al., 2023; Chen & Kae, 2019), generating food images (Papadopoulos et al., 2019; Han et al., 2020b;a), and generating image from object layout (Arad Hudson & Zitnick, 2021; Zheng et al., 2023; Zhao et al., 2019; Yang et al., 2022; Zeng et al., 2023). Meanwhile, compositing the concepts beyond objects for image generation has received wide attention. (Du & Kaelbling, 2024; Liu et al., 2022; 2023; Du et al., 2020; 2021) learn to generate images with different concepts by compositing smaller models trained on different objects. For instance, (Liu et al., 2023) learns to composite styles from Van Gogh and Monet's paintings with pretrained text-to-image diffusion models. Instead of using different models for composition, (Abdal et al., 2021; Nie et al., 2021) use the attribute labels as condition to learn a single generative model and generate new images by changing the input label. However, previous methods treats the label (e.g., text or binary attribute) as concept and may leads to unwanted changes when they are expected to represent different meanings. For example, in human face editing, the label *5 o'clock Ssadow* is causally related with the labels *male* and *young*. Thus the label also contains information about the gender and age. If we ignore such difference between label and concept, adding *5 o'clock shadow* to a young girl face could be impossible (see Figure 1) or makes the girl looks like a male and older (see Figure 6).

In this paper, we present a generative model to learn concepts from data and attribute labels. Given binary attribute label, we aim to disentangle the true concept from correlated information such that we can change only this concept to mkae it controllable. To this end, we formulate the *minimal change principle* where the changes brought by each concept should be minimal, i.e., the concept does not contain information about other concepts. According to this principle, we make an assumption about the data generating process where the latent variables (concepts) are generated in a compositional way. It means that the concept is activated if label is 1 or deactivated if label is 0. To minimize the influence of each concept, we introduce a binary mask to find the dimension of each concept automatically. Then we show that adding the sparsity achieves better compositional performance. In addition, the labels could be dependent (spuriously correlated, in some contexts) and we propose a causal conditioning method to handle it in an efficient way. Specifically, we perform causal discovery on the labels and then condition the child variable on the parent variables to isolate the influence from parent variables when we only want to change child variables. With these two important modules, we show that the concepts are identifiable and superior performances demonstrate the efficacy of our method and validates our theory.

Our proposed method has several theoretical and practical contributions:

(i)      We formulate the minimal change principle and present an empirical method to learn concepts for compositional image generation guided by this principle.

(ii)      We propose a way to disentangle the concept from causally related attribute labels by factorizing the information.

(iii)      We prove that the true concepts could be recovered in our method under some assumptions.

(iv)      We empirically demonstrate that our method achieves better generation and editing results. The significant gain over the baseline methods on different tasks demonstrate the effectiveness of our technique.

## 2    RELATED WORK

**Causal Representation Learning, Concept Learning, and Compositional Image Generation**
The goal of causal representation learning (CRL) is to reconstruct the true generation process of the data (Schölkopf et al., 2021). It can be viewed as a combination of disentanglement (Hyvärinen et al., 2023), representation learning (Bengio et al., 2013), and causal discovery (Spirtes et al., 2001; Glymour et al., 2019). Identifiability plays an important role in CRL. When the distributions of the estimation model and real data are matched, the learned latent variable is shown to be an invertible transformation of the true factor, i.e., the estimation only contains the information about the true factor (we call the variable *identifiable*). Learning concepts for compositional image generation is highly related to CRL as the identifiability guarantees many desirable properties of the estimation model, such as robustness to outliers. For instance, CRL has been shown to be effective in many downstream classification tasks (Brehmer et al., 2022; Mitrovic et al., 2020; Wang et al., 2022b; Lu et al., 2021). Recently, Rajendran et al. (2024) show that CRL can improve large language models.

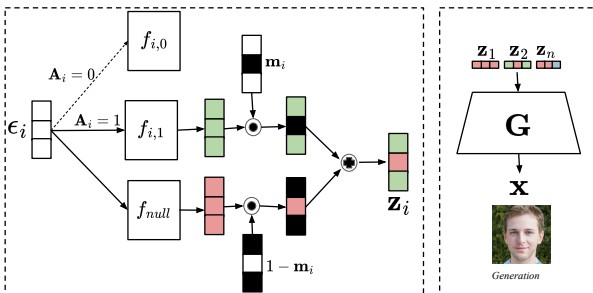

Figure 2: The image generation process of our estimation model. We transform the Gaussian noise $\epsilon_i$ according to the label $\mathbf{A}_i$. To reduce the influence of each concept, we learn a binary mask $\mathbf{m}_i$ to decide which elements should be transformed. Finally, we concatenate latents into the generator $G$.

In this paper, we present a framework for identifiable concepts and explore how CRL benefits the compositional image generation. Learning concepts for compositional image generation is gaining wide attention recently. Du & Kaelbling (2024); Liu et al. (2022; 2023); Du et al. (2020; 2021) assume that concepts are represented by the labels or texts. They further assume that the concepts are independent and thus their joint distribution can be factorized and modeled by separate smaller models. Compared to their methods, this focuses on the case where attribute labels contain redundant information about the concepts and how we can drop it for better compositional image generation.

**Attribute Manipulation and Conditional GAN** Human attributes manipulation is an important topic. StyleGAN-based approaches are roughly divided into three steps: 1. inferring the latent code of real images with pretrained StyleGAN generator using optimization-based methods (Karras et al., 2020b) or encoder-based methods (Abdal et al., 2019; Richardson et al., 2021; Tov et al., 2021; Wang et al., 2022a; Pehlivan et al., 2023); 2. training additional network to generate the inferred latent codes with attribute labels (Abdal et al., 2019; Li et al., 2023; Huang et al., 2023a; Suwała et al., 2024; Nie et al., 2021) or learning editing directions with attribution labels (Shen et al., 2020b;a; Wu et al., 2021; Ling et al., 2021); 3. generating a new latent code by changing the attribute label and feed it into the pretrained StyleGAN generator. Some attempts have been made with text-guided image manipulation (Patashnik et al., 2021; Wei et al., 2023; Huang et al., 2023b). At the same time, many methods formulate this problem as a multi-domain image-to-image translation problem and apply different regularizations (Choi et al., 2018; Xiao et al., 2017; 2018; Kim et al., 2024). There are also some causality inspired methods in image generation, such as CausalGAN (Kocaoglu et al., 2017). However, CausalGAN has to model the generation as a causal graph and may not be able to scale up when the number of attributes is large. We present more causal-aware generative models in the appendix A. Conditional GAN (Mirza, 2014) is closely related to the attribute manipulation and its goal is to generate high-quality images from given classes. AugGAN (Hou et al., 2024) encourages the discriminator to predict the augmentation parameters to avoid the negative impact from data augmentation.

## 3 COMPOSITIONAL IMAGE GENERATION AND EDITING

Given a set of images $\mathbf{x}$ and their corresponding attribute labels $\{\mathbf{A}_i\}_{i=1}^n$ ($n$ is the number of labels for each image), our goal is to learn concepts for these attribute label such that the learned model generates images in a compositional way. We expect that our model is able to generate combinations that never appear in the training set. We present the minimal change principle in Section 3.1, the causal conditioning method to handle dependent labels in Section 3.2, the empirical estimation in Section 3.3, and the real-image editing method in Section 3.4. Finally, we provide identifiability guarantees for our learned concepts in Section 3.5. The whole computation flow of our method is present in Figure 2.

### 3.1 MINIMAL CHANGE PRINCIPLE AND IMAGE GENERATION PROCESS

To achieve compositional image generation and editing, we first analyze the nature of compositional image generation. It would be desirable to have each label control a specific part of the generation. If the influence of each label is large, it will bring additional distortions and can lead to failure of

composition. Therefore, we formulate the minimal change principle as *the influence brought by each true concept should be minimal such that it brings only necessary changes while adding any tiny influence will causes unwanted changes*. In light of this principle, we make an assumption where an image $x$ is composed of $n$ concepts $\mathbf{z}^*$ such as color and shape. Formally, we assume that

$$\mathbf{x} = g(\mathbf{z}_1^*, \dots, \mathbf{z}_n^*, \mathbf{z}_c^*), \tag{1}$$

where $\mathbf{z}_i^* \in \mathbb{R}^{d_i}$ denotes the $i$-th concept with dimension $d_i$ and $\mathbf{z}_c^*$ is the remaining part. We also write $\mathbf{Z}^* = (\mathbf{z}_1^*, \dots, \mathbf{z}_n^*, \mathbf{z}_c^*)$. The distribution of each concept $\mathbf{z}_i^*$ is decided by a binary mask $\mathbf{A}_i \in \{0, 1\}$ as

$$\mathbf{z}_i^* = f_{i,1}^*(\epsilon_i^*) \odot \mathbf{A}_i + f_{i,0}^*(\epsilon_i^*) \odot (1 - \mathbf{A}_i), \tag{2}$$

where $\epsilon_i^* \in \mathbb{R}^{d_i}$ is sampled from a prior distribution, e.g., $\mathcal{N}(0, I)$, $f_{i,1}^*$ denotes the corresponding activating function, and $f_{i,0}^*$ denotes the deactivating function. An example would be that $\mathbf{m}_i$ denotes whether the man has a mustache. If the man has mustache, i.e., $\mathbf{m}_i = 1$, the function of the mustache concept $f_i^*$ would be activated and we have $\mathbf{z}_i^* = f_{i,1}^*(\epsilon_i^*)$. The generation process is designed to be compositional and reduces the influence of each concept. In other words, changing the label only leads to changes in the corresponding dimensions of the concept. In additional, *the dimensionality of each concept is constrained to be minimal under the constraint that the information of the concept is rich enough to explain the image*, and we show that how we can achieve it in an efficient way in the following sections.

## 3.2 CAUSAL CONDITIONING FOR DEPENDENT LABELS

The minimal change principle guides us to restrict the information from attribute label. However, the real-world labels could be heavily dependent from each other; if such dependence is not properly address, it would lead to unsatisfactory disentanglement even if minimal change has been enforced. For example, *old* (O) and *eyeglasses* (E) are usually highly correlated. When one tries to activate the eyeglasses concept, it contains the information from age. Consequently, activating the eyeglasses concept alone could result in additional age-related distortions (see Figure 3). To disentangle these concepts, we resort to a causal perspective and provide a natural and effective solution.

Take the age-eyeglasses as an example, we have causal model $O \rightarrow E \leftarrow \varepsilon$, where $\varepsilon$ is an unobserved exogenous random variable as shown in Figure 4(a). Consequently, the eyeglasses node contains information about the age. To change eyeglasses without changing the age influence, we can consider the latent variable $\varepsilon$ because of $E = f_E(O, \varepsilon)$, the structure causal model to describe the causal process (Pearl, 2009; Spirtes et al., 2001). Accordingly, we propose to factorize the dependent label information in Table 4(b). Since the original column $E$ contains the information about age, we can build two new columns $E|O = 1$ and $E|O = 0$ to represent the specific information from $\varepsilon$. Without knowing whether a person is young or old, we cannot decide whether that person is wearing eyeglasses with only two columns. We can infer whether the person wearing eyeglasses through combining the information from column $O$ and columns $E|O = 1$ and $E|O = 0$.

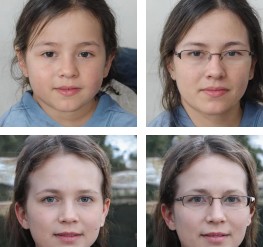

Figure 3: Examples of activating eyeglasses. Top: no causal. Bottom: with causal conditioning.

Therefore, given attribute labels, we first run causal discovery algorithms to discover the causal relationships among the attributes, e.g., PC (Spirtes et al., 2001). Depending on the chosen causal discovery method, in the output, some edges may not be oriented, for instance, if PC is used. In that case, we further use our background knowledge to determine the directions for those edges. Then for each attribute, we can factorize the information by conditioning on its parent attributes like Figure 4(c). Then we replace the given labels with this transformed labels to train our generative models. Such a transformation could be done in several seconds and easily scale up with the number of attribute labels. We present more details and visualizations about causal conditioning in the Appendix D.

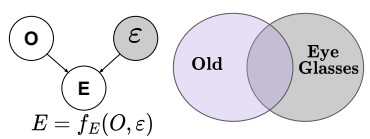

$$E = f_E(O, \varepsilon)$$

(a) Old causes the eyeglasses and the shared information leads to additional changes when we only want to change eyeglasses.

| $O$ | $E$ |
|---|---|
| 1 | 1 |
| 1 | 0 |
| 0 | 1 |
| 0 | 0 |

(b) Given attribute labels. O denotes old and E denotes eyeglasses.

| | $\varepsilon$ | |
|---|---|---|
| $O$ | $E\mid O=1$ | $E\mid O=0$ |
| 1 | 1 | 0 |
| 1 | 0 | 0 |
| 0 | 0 | 1 |
| 0 | 0 | 0 |

(c) Transformed labels. label $O$ no longer share information with the rest two columns.

Figure 4: Separate the information of the child variable *Eyeglasess* from its parent *Old*.

## 3.3 ESTIMATION

We are now ready to present our implementation for estimating the true concepts and using them. We first build our network to follow the generation process. For estimation of the $i$th concept $\mathbf{z}_i$, we have

$$\mathbf{z}_i = f_{i,1}(\epsilon_i) \odot \mathbf{A}_i + f_{i,0}(\epsilon_i) \odot (1 - \mathbf{A}_i), \tag{3}$$

where $\epsilon_i \sim P_\epsilon$ and we also take $P_{\epsilon_i} = \mathcal{N}(0, I)$, where $f_{i,1}$ and $f_{i,0}$ are constructed as multi-layer perceptron (MLP). After obtaining the concepts $\{\mathbf{z}_i\}$, we feed them into a generator $G$ as

$$\mathbf{x}^{\text{fake}} = G(\mathbf{z}_1, \ldots, \mathbf{z}_n, \mathbf{z}_c) \tag{4}$$

and perform adversarial training between the discriminator $D$ to match the marginal distributions between generated images and real images:

$$\mathcal{L}_{\text{adv}} = \mathbb{E}_{x,a} \log D(x|a) + \mathbb{E}[1 - \log D(x^{\text{fake}}|a)], \tag{5}$$

where $x$ and $a$ are sampled images and attribute labels and the discriminator $D$ is trained to maximize the loss while the generator is trained to minimize the loss.

However, concepts have different meanings; some are simple while some others are complex. They are expected to correspond to different dimensions in the representation. To recover the true minimal dimension for each concept, we propose to employ additional learnable mask $\mathbf{m}_i$ for $i$-th concept as

$$\mathbf{z}_i^{\text{deact}} = f_{i,1}(\epsilon_i), \quad \mathbf{z}_i^{\text{act}} = f_{i,0}(\epsilon_i), \quad \mathbf{z}_i^{\text{null}} = f_{\text{null}}(\epsilon_i) \tag{6}$$

$$\mathbf{z}_i = \mathbf{m}_i \odot (\mathbf{z}_i^{\text{act}} \odot \mathbf{A}_i + \mathbf{z}_i^{\text{deact}} \odot (1 - \mathbf{A}_i)) + (1 - \mathbf{m}_i) \odot \mathbf{z}_i^{\text{null}},$$

where $f_{\text{null}}$ is a shared network across different attributes, i.e., it does not contain any information about concept. The mask $\mathbf{m}_i$ can be set to a fixed larger dimension and we apply an $l_1$ sparsity loss to select the important and necessary elements:

$$\mathcal{L}_{\text{sparsity}} = \sum_{i=1}^{n} \|\mathbf{m}_i\|_1. \tag{7}$$

Let $\lambda_{\text{sparsity}}$ be a hyperparameter controlling the sparsity of learned mask. Our full objective is then

$$\mathcal{L}_{\text{full}} = \mathcal{L}_{\text{adv}} + \lambda_{\text{sparsity}} \mathcal{L}_{\text{sparsity}}, \tag{8}$$

## 3.4 COMPOSITIONAL IMAGE EDITING

In previous sections, we can change the binary labels $\mathbf{A}_i$ to achieve controllable image generation. However, it still remains challenging to edit the real images. A straight-forward way is to project image $\mathbf{x}$ into our feature space $\mathcal{Z}$ (or $\mathcal{W}$ space in StyleADA (Karras et al., 2020a)). However, to achieve editing, after inversion, we still have to train an additional network to incorporate the attribute label information like Abdal et al. (2019); Pehlivan et al. (2023). In this section, we choose to invert an image $\mathbf{x}$ to the input space rather than latent space $\mathcal{Z}$, i.e., we have to recover the noise variables $\{\epsilon_i\}$. After recovering the noise $\epsilon_i$, we can choose to activate or deactivate the concept by feeding different label $\mathbf{A}_i$. This utilizes the label information during inversion process unlike Karras et al. (2020a) while avoiding the two-stage training for editing like Pehlivan et al. (2023).

We first initialize the noise variables $\{\hat{\epsilon}_i\}$ from Gaussian distribution $\mathcal{N}(0, I)$. Then with the attribute label of the image $\{\mathbf{A}_i\}$ (could be easily obtained from a pre-trained classifier), we obtain the latent variable $\{\hat{\mathbf{z}}_i\}$ following eq. 3. Finally, we generate the reconstructed image $\hat{\mathbf{x}} = G(\{\hat{\mathbf{z}}_i\})$. We optimize the noise variable $\{\epsilon_i\}$ with following loss function:

$$\mathcal{L}_{\text{inversion}} = \lambda_{\text{recon}}\mathcal{L}_{\text{recon}} + \lambda_{\text{reg}}\mathcal{L}_{reg} + \lambda_{\text{lpips}}\mathcal{L}_{\text{lpips}} + \lambda_{\text{kl}}\mathcal{L}_{\text{kl}}, \tag{9}$$

where $\mathcal{L}_{\text{recon}} = \|\mathbf{x} - \hat{\mathbf{x}}\|_2^2$, and $\mathcal{L}_{\text{lpips}} = \|(\text{VGG}(\mathbf{x}) - \text{VGG}(\hat{\mathbf{x}}))\|$ with the pretrained VGG network. The second term $\mathcal{L}_{\text{reg}}$ is the noise regularization used in StyleADA (Karras et al., 2020a) to regularize the noise maps in the generator (we present the details in appendix). The last term $\mathcal{L}_{\text{kl}} = KL(p_{\hat{\epsilon}_i}, \mathcal{N}(0, I))$ is to regularize the noise variables staying in the Gaussian space since our model is trained with Gaussian noise input. We find that this term leads to better result for non-curated images. The coefficients $\lambda_{\text{recon}}, \lambda_{\text{reg}}, \lambda_{\text{lpips}}, \lambda_{\text{kl}}$ balance the effects from different loss terms. In most experiments, we set $\lambda_{\text{recon}} = 1$, $\lambda_{\text{reg}} = 10000$, $\lambda_{\text{lpips}} = 1$, and $\lambda_{\text{kl}} = \{0, 0.001\}$.

After training, we can obtain the corresponding noise variables that well represents the real images $\mathbf{x}$. Then we can edit the images by fixing the noise variable and change the labels $\mathbf{A}_i$ to activate or deactivate a concept to achieve composite image editing.

### 3.5 Identifiability Guarantee for the Concepts

In this section, we present identifiability guarantee for learning the underlying concepts. Specifically, we show that our estimation model recovers the true latent variables $\mathbf{z}_i^*$ up to invertible transformation under some conditions. The key idea is to leverage sufficient variability across activating and deactivating functions to identify the concepts. The proof of the following result is provided in Appendix C.

**Theorem 1** (Identifying concepts with a generative model). *Consider the data generating process defined in Eqs. (1) and (2) that satisfy the following assumptions:*

- *A1 (Smooth and positive density): The density function $p_\epsilon$ is continuously differentiable and positive everywhere.*

- *A2 (Diffeomorphism): The generating function $g$, activation function $f_{i,1}$, and deactivating function $f_{i,0}$ are $\mathcal{C}^2$-diffeomorphisms onto their corresponding images.*

- *A3 (Concept regularity): The activating function $f_{i,1}$ differs sufficiently from the deactivating function $f_{i,0}$, and each concept is of dimension one. Specifically, for $i = 1, \ldots, n$ and every $\epsilon_i^*$, we have*

$$\frac{\partial}{\partial \epsilon_i^*}\left(\frac{p_{\epsilon_i^*}}{f'_{i,1}}\right) \neq \frac{\partial}{\partial \epsilon_i^*}\left(\frac{p_{\epsilon_i^*}}{f'_{i,0}}\right) \quad \text{or, equivalently,} \quad \left(\frac{p_{\epsilon_i^*}}{f'_{i,1}}\right)' \neq \left(\frac{p_{\epsilon_i^*}}{f'_{i,0}}\right)'.$$

- *A4 (Concept variability): For each concept $\mathbf{z}_i$, there exist two attribute labels, denoted as $\mathbf{A}^{(k)}$ and $\mathbf{A}^{(l)}$, that differ only in the $i$-th entry.*

*Then, for a generative model that assumes the same generative process, satisfies the assumptions above, and matches the data distribution, i.e., $p_{G(\mathbf{Z})} = p_\mathbf{x}$, it identifies the true latent variables $\mathbf{z}_i^*$ up to invertible transformation for each attribute $\mathbf{A}_i$.*

Assumptions A1 and A2 are standard assumptions for identifiability in representation learning (Hyvarinen et al., 2019; Hyvärinen et al., 2023), ensuring that the functions are well-behaved. Assumption A3 requires the activating function to differ sufficiently from the deactivating function, which is analogous to the sufficient change assumption in disentangled (Hyvarinen et al., 2019; Hyvärinen et al., 2023) or causal representation learning (Schölkopf et al., 2021; Zhang et al., 2024) that requires the data distribution of latent variables to vary sufficiently. In our setting, such an assumption is needed because, intuitively speaking, it is in general not possible to identify the concepts if activating and deactivating the concepts lead to highly similar data distributions. Furthermore, Assumption A4 imposes a certain type of variability on the attribute labels, which is analogous to humans learning one concept at a time.

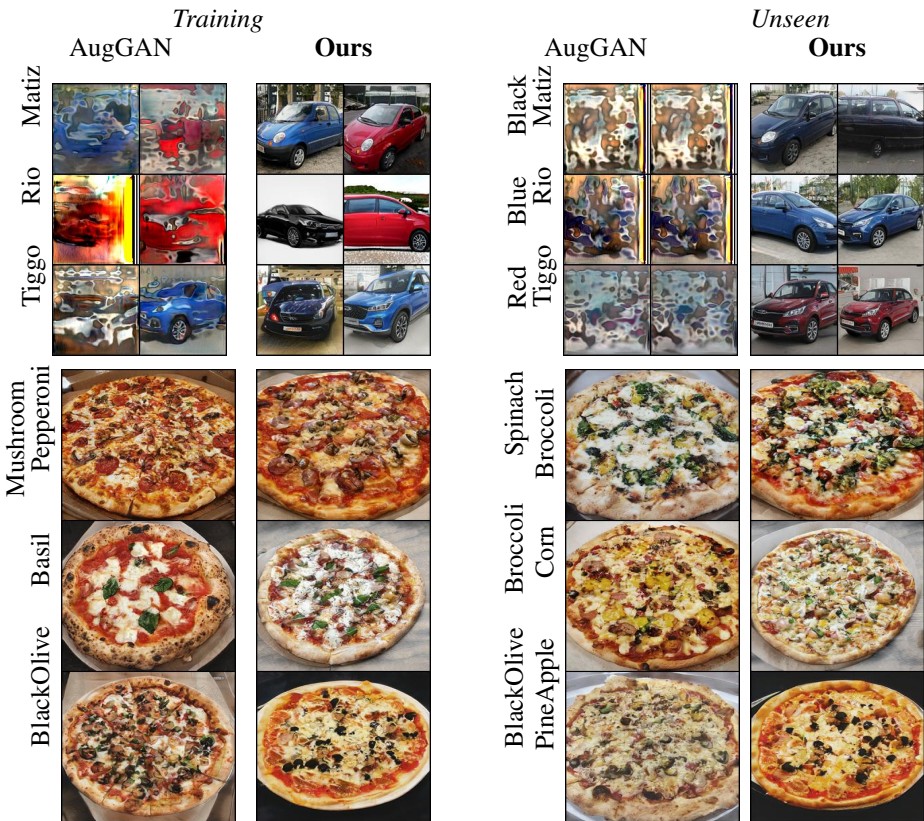

Figure 5: The generations on Car and Pizza dataset. The SOTA method, AugGAN (Hou et al., 2024), still fails in generating car images with limited data. It also unable to generate images that align with pizza labels, e.g., the black olive is generated with broccoli.

| Datasets | Metrics | StyleADA (Karras et al., 2020a) | AugGAN (Hou et al., 2024) | Ours |
|---|---|---|---|---|
| MNIST4 | In-FID ↓ | 106.39 | 97.37 | **5.27** |
| | Out-FID ↓ | 155.07 | 147.83 | **43.55** |
| Car9 | In-FID ↓ | 139.79 | 332.71 | **10.46** |
| | Out-FID ↓ | 202.18 | 335.72 | **20.38** |
| Pizza13 | FID ↓ | 4.15 | 4.75 | **4.12** |
| | Out-FID ↓ | 9.45 | 9.98 | **9.15** |

Table 1: The generation results on independent attribute datasets. The In-FID measures the divergence between the generated and training images. To compute Out-FID, we generate images with unseen or rare labels and measure the divergence between the generated and test images.

## 4 EXPERIMENTS

### 4.1 SETUP

**Datasets**. We consider two groups of experiments to investigate the performance of our proposed method. The first consists of experiments on datasets with independent concepts. The second focuses on experiments where there exists dependence among different attribute labels.

(i) Independent attributes. We use MNIST4, Car9 and Pizza13 datasets. MNIST4 contains 4 combinations with green(red) foreground and white(black) background. Each combination contains 500 images. As for the car dataset, it contains 3 car types and 3 colors. Each combination contains around 500 images with resolution 128×128. As for Pizza13 dataset, we use the pizza dataset (Papadopoulos et al., 2019) which contains 13 ingredients of pizza images with resolution $256 \times 256$.

| GPT-4o | Meta AI | Stable 3 | FaceDiffusion | Ours |

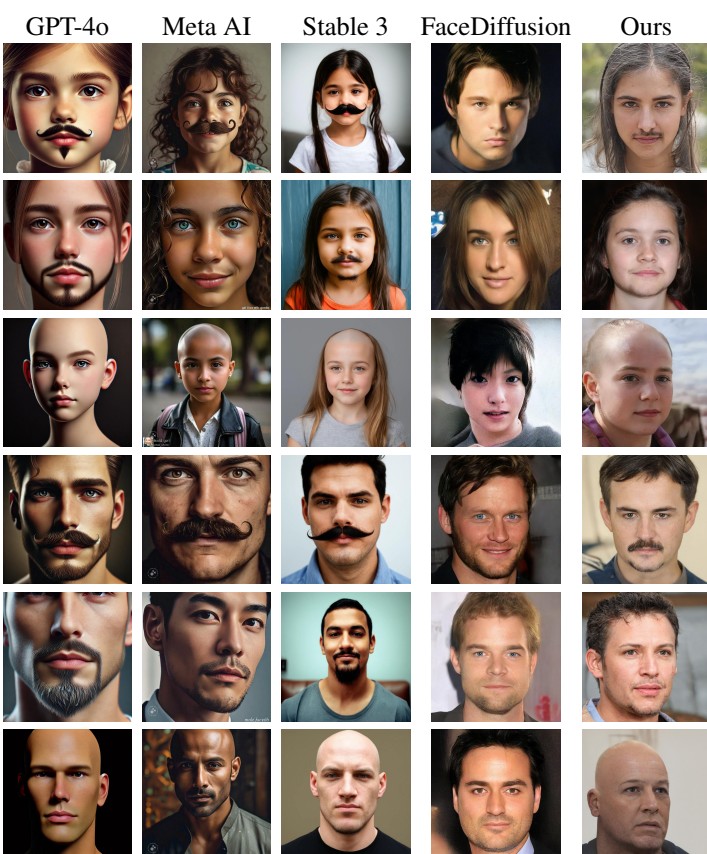

Figure 6: Comparisons of generation with different conditions: *girl with mustache, girl with goatee, bald girl, male with mustache, male with goatee and bald male*. When trying to generate some non-existent combinations of concepts, existing AIs tends to generate some *fake* concepts, such as the cartoon mustache by Stable Diffusion 3 (first row, third column).

(ii) Dependent attributes. We use FFHQ dataset (Karras et al., 2019). We use a pre-trained CELEBA classifier to obtain attributes on the 70,000 images. We use 37 attributes of the human face. The dataset is trained at resolution 512×512.

**Implementation**. We develop our method based on the StyleGAN2-ADA code (Karras et al., 2020a). We provide our code in the supplementary material. The details are present in the appendix.

| Method | FID ↓ | KID×100 ↓ | Precision ↑ | Recall ↑ |
|---|---|---|---|---|
| Ours ($\lambda = 0$) | 5.68 | 0.13 | 0.66 | 0.43 |
| +Sparsity ($\lambda = 0.1$) | 4.42 | 0.13 | 0.65 | 0.47 |
| +Causal Conditioning | 3.94 | 0.10 | 0.66 | 0.45 |

Table 2: Ablation of generation results on the proposed components on FFHQ dataset.

## 4.2 RESULTS ON DATASETS WITH INDEPENDENT ATTRIBUTES

We present the quantitative result of generating images in Table 1. The combinations have been seen by all models and we observe that our method achieve the best performance across different datasets. It is worth noting that the baseline methods, StyleADA (Karras et al., 2020a) and AugGAN (Hou et al., 2024), suffer the mode collapse on the MNIST4 and Car9 datasets. A main reason might be that the number of training images on these two datasets is very limited (500 images per combination). With our minimal change principle, our method is data-efficient in learning representations of the concepts. For instance, although there are only 500 black matiz car images, there are $500 \times 2$ matiz images and $500 \times 2$ black car images. As shown in Figure 5, our method learns to generate

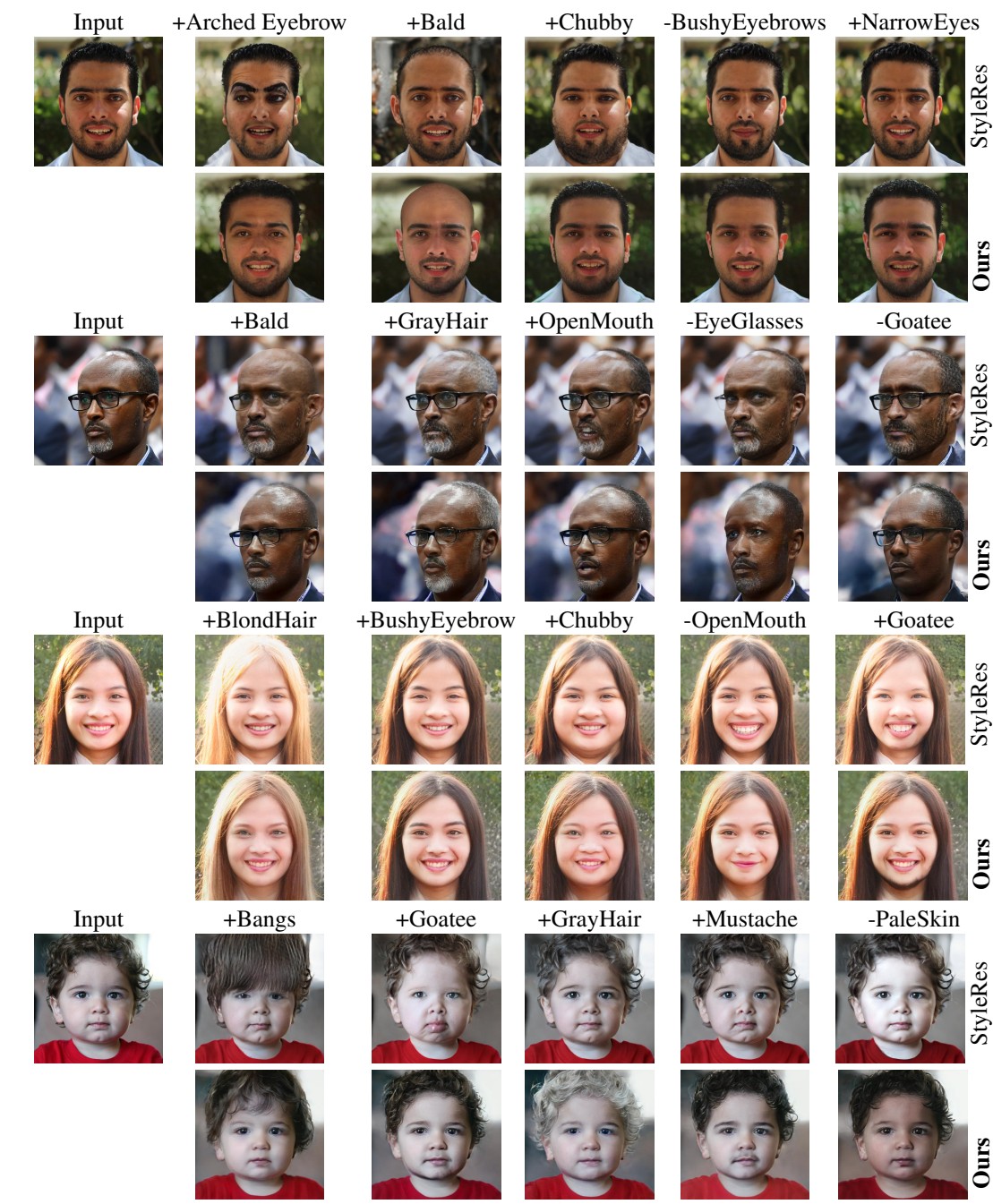

Figure 7: Comparisons on the real image editing task. SOTA method– StyleRes(Pehlivan et al., 2023) achieves good performance in common combinations, such as chubby faces but fails in other cases, e.g., adding goatee to a baby face (see the second last row). Sometimes, it also confuses between activating and deactivating a condition, e.g., closed mouth rather than a opening mouth.

different cars and pizza according to the given conditions while the strong baseline AugGAN (Hou et al., 2024) fails. It is encouraging that our method achieves satisfactory Out-FID results on different datasets although the labels have never or rarely been seen by the model. As shown in the Figure 5, our method learns to generate different car and pizza images. Importantly, our model follows the conditions while the baseline method AugGAN (Hou et al., 2024) fails to consider the ingredients as concepts. For instance, although we only want to activate black-olive concept in pizza, AugGAN still generates brocolli in the output images.

| Task | FaceDiffusion | Ours |
|---|---|---|
| Girl-Mustache | 0.215 | **0.273** |
| Girl-Goatee | 0.217 | **0.230** |
| Girl-Bald | 0.219 | **0.271** |
| Man-Mustache | 0.221 | **0.272** |
| Man-Goatee | 0.221 | **0.224** |
| Man-Bald | 0.221 | **0.271** |

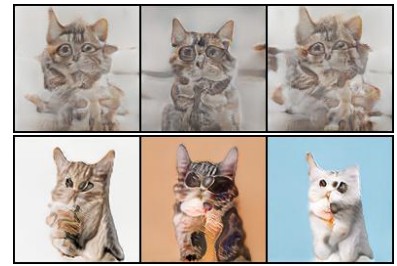

Figure 8: We measure the generation performance on human face with seen and unseen labels, e.g., generating a girl face with a mustache. Since existing images of the given conditions can be very rare or non-existent, we use CLIP to measure the alignment between the generation and the given conditions.

Figure 9: Failure case: *cat eating ice cream* with 50 training images. Top row: AugGAN, bottom row: Ours.

### 4.3 RESULTS ON DATASETS WITH CORRELATED ATTRIBUTES

**Generation results** We compare with the SOTA method in face generation, FaceDiffusion (Huang et al., 2023b). To test the ability of compositing concepts, we ask the models to generate images under 6 conditions: *girl with mustache*, *girl with goatee*, *bald girl*, *male with mustache*, *male with goatee*, and *bald male*. The last three conditions are common in human face images while the first three conditions are rare or do not exist. As shown in Table 8, our method achieves much better performances across the 6 tasks. We also provide the comparisons of generated images between the strongest AI models, including GPT-4o, Meta AI, and Stable Diffusion3 in Figure 6. We observe that the foundational models still struggle in generating realistic images of the unseen combinations. For instance, GPT-4o and Stable Diffusion3 generate cartoon mustaches on the girl's face, rendering the whole image unnatural. In contrast, our method learns to add mustache to girl and male faces. This demonstrate the advantage of learning identifiable concept representations from images.

**Editing results** To testify the editing ability of our approach, we compare our method with the SOTA face editing method,StyleRes (Pehlivan et al., 2023). StyleRes is trained to inverse the StyleGAN by projecting images into a higher-dimensional space and leverages second-stage editing techniques such as InterfaceGAN (Shen et al., 2020b) and StyleClip (Patashnik et al., 2021) to achieve image editing. As shown by the editing results in Figure 7, the strong baseline StyleRes achieves very good reconstruction since it learns to reconstruct the image in the high-dimensional space. However, very high-dimensional features may also lead to worse editing results, because of potential redundancy. For instance, it fails to narrow the eyes in the first row and remove the goatee in the third row.

**Ablation** To testify the effects of our proposed minimal change principle and the causal conditioning, we perform ablation on the FFHQ dataset. As shown in Table 2, the model achieves the worst performance if no sparsity (to achieve minimal changes) is added. With no sparsity, the latent could be very high-dimensional and contains redundant information and leads to additional changes in the output images. With the proposed causal conditioning, the model achieves the best performance.

## 5 LIMITATIONS, DISCUSSION, AND CONCLUSION

Although we have shown that our method has achieved best performance across different compositional image generation and editing tasks, our method still suffers from following limitations: 1) Difficulty in generating complex images. We build our model based on StyleADA, which is known to be challenging to generate non-curated images (Sauer et al., 2022). We have explored learning concepts with cat and icecream with 50 training images. Unfortunately, neither our method nor the baseline AugGAN succeeded, as shown in Figure 9. 2) Lacking multi-modal control. Our model currently only supports attribute labels to control the generation process, which is limited compared to foundational models like GPT-4o. Adding text and mask control can enlarge the concept space but could be more challenging, and we leave the above two problems as future work.

In this paper, we present a concept-learning framework for compositional image generation and editing. To learn the concepts, we introduce the minimal change principle to restrict the influence of each concept. In addition, we also present a causal conditioning method to address the dependent attribute label problem. We have shown that our method achieves the best performances across different datasets and tasks. In summary, the proposed framework is more data-efficient, controllable, and interpretable.

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

## A    RELATED WORK

**Causally-aware Generative Models** CausalGAN (Kocaoglu et al., 2017) proposes to build a causal generative following the causal graph. For example, if $X \rightarrow Z \leftarrow Y$, then it employs three neural networks for each node and the input of neural network of $Z$ consists of the outputs of neural network for node $X$ and node $Y$. CausalVAE (Yang et al., 2021) assumes that there exists underlying causal structure among the latent variables and add a causal layer to learn such information. Wen et al. (2022) employs CausalGAN for tabular data geverife while supporting part of the causal graph. CGNN (Goudet et al., 2018) learns causal graph and the functions by minimizing the divergence between the generated data and real data. DECAF (Van Breugel et al., 2021) reconstructs each variable with its parents as conditioning and generates fair synthetic data. Moraffah et al. (2020) assumes the Structural Causal Model (SCM) is linear and proposes to learn the causal graph with GAN. CNF (Javaloy et al., 2024) recovers the causal model with normaling flow given causal ordering information. CGN (Sauer & Geiger, 2021) assumes that images are generated by four components: shape, texture, background and composer and train a conditional GAN with corresponding labels. CAGE (Bose et al., 2022) examines the causal relationship between a pair of variables using potential outcome framework and generates counterfactual images.

**Nonlinear ICA** Nonlinear ICA is a challenging ill-posed problem because the latent variables are generally not identifiable without any assumptions (Hyvärinen & Pajunen, 1999). Existing works resolve this issue by leveraging sufficient variability on the distribution of latent variables to obtain identifiability, where the distributions are indicated by auxiliary variables such as time indices and domain indices (Hyvarinen & Morioka, 2016; 2017; Hyvarinen et al., 2019; Khemakhem et al., 2020). Another line of works impose restrictions on the mixing function including certain function classes (Hyvärinen & Pajunen, 1999; Taleb & Jutten, 1999; Gresele et al., 2021; Buchholz et al., 2022) and sparse mixing function (Zheng et al., 2022).

## B    CAUSAL DISCOVERY

We perform causal discovery on the FFHQ attribute labels and the causal graph is shown in Figure 10.

## C    PROOF OF IDENTIFIABILITY THEORY

**Theorem 1** (Identifying concepts with a generative model). *Consider the data generating process defined in Eqs. (1) and (2) that satisfy the following assumptions:*

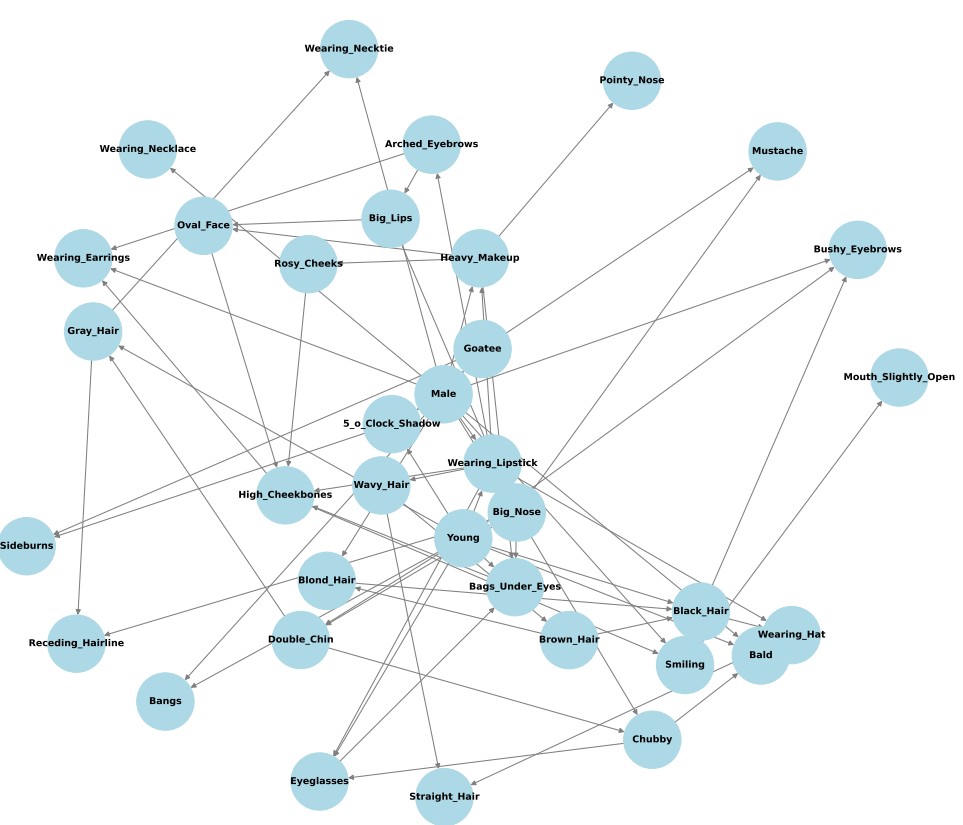

Figure 10: Causal discovery result on the FFHQ attributes labels using algorithm in (Andrews et al., 2023). By conditioning on its parent variables, we disentangle the concept-related information from the correlated attribute labels.

- *A1 (Smooth and positive density): The density function $p_\epsilon$ is continuously differentiable and positive everywhere.*

- *A2 (Diffeomorphism): The generating function $g$, activation function $f_{i,1}$, and deactivating function $f_{i,0}$ are $C^2$-diffeomorphisms onto their corresponding images.*

- *A3 (Concept regularity): The activating function $f_{i,1}$ differs sufficiently from the deactivating function $f_{i,0}$, and each concept is of dimension one. Specifically, for $i = 1, \ldots, n$ and every $\epsilon_i^*$, we have*

$$\frac{\partial}{\partial \epsilon_i^*}\left(\frac{p_{\epsilon_i^*}}{f_{i,1}'}\right) \neq \frac{\partial}{\partial \epsilon_i^*}\left(\frac{p_{\epsilon_i^*}}{f_{i,0}'}\right) \quad \text{or, equivalently,} \quad \left(\frac{p_{\epsilon_i^*}}{f_{i,1}'}\right)' \neq \left(\frac{p_{\epsilon_i^*}}{f_{i,0}'}\right)'.$$

- *A4 (Concept variability): For each concept $\mathbf{z}_i$, there exist two attribute labels, denoted as $\mathbf{A}^{(k)}$ and $\mathbf{A}^{(l)}$, that differ only in the $i$-th entry.*

*Then, for a generative model that assumes the same generative process, satisfies the assumptions above, and matches the data distribution, i.e., $p_{G(\mathbf{Z})} = p_{\mathbf{x}}$, it identifies the true latent variables $\mathbf{z}_i^*$ up to invertible transformation for each attribute $\mathbf{A}_i$.*

*Proof.* To ligthen the notation, let $\mathbf{Z}^* = (\mathbf{z}_1^*, \ldots, \mathbf{z}_n^*)$ and $\mathbf{Z} = (\mathbf{z}_1, \ldots, \mathbf{z}_n)$. We then have $\mathbf{x}^* = g(\mathbf{Z}^*)$ and $\mathbf{x} = G(\mathbf{Z})$. Applying the change-of-variable formula with matched data distribution yields

$$p_{\mathbf{x}} = p_{\mathbf{x}^*} \implies p_{G(\mathbf{Z})} = p_{g(\mathbf{Z}^*)} \implies p_{g^{-1} \circ G(\mathbf{Z})} \operatorname{vol} J_{g^{-1}} = p_{\mathbf{Z}^*} \operatorname{vol} J_{g^{-1}} \implies p_{h(\mathbf{Z})} = p_{\mathbf{Z}^*}.$$

Here, $J_{g^{-1}}$ denotes the Jacobian matrix of function $g^{-1}$ and $h = g^{-1} \circ G$ is the function from $\mathbf{Z}^*$ to $\mathbf{Z}$. Applying the change-of-variable formula again, we have

$$p(\mathbf{Z}) = p(\mathbf{Z}^*)|\det J_h|.$$

where $J_h$ denotes the Jacobian matrix of function $h$. Since $\mathbf{Z}$ and $\mathbf{Z}^*$ consist of independent latent variables, we obtain

$$p(\mathbf{z}_c)\prod_{i=1}^{n} p(\mathbf{z}_i) = p(\mathbf{z}_c^*)\prod_{i=1}^{n} p(\mathbf{z}_i^*)|\det J_h|. \tag{10}$$

Consider latent variable $\mathbf{z}_i$ where $i = 1, \ldots, n$. By Assumption A4, there exist two attribute labels, denoted as $\mathbf{A}^{(k)}$ and $\mathbf{A}^{(l)}$, that differ only in the $i$-th entry. Without loss of generality, suppose $\mathbf{A}_i^{(k)} = 1$ and $\mathbf{A}_i^{(l)} = 0$; if this is not the case, we can swap $\mathbf{A}^{(k)}$ and $\mathbf{A}^{(l)}$. Denote the corresponding density functions for these two attribute labels by $p^{(k)}(\mathbf{Z}), p^{(k)}(\mathbf{Z}^*)$ and $p^{(l)}(\mathbf{Z}), p^{(l)}(\mathbf{Z}^*)$, respectively. Substituting these density functions into Eq. (10) implies

$$p^{(k)}(\mathbf{z}_c)\prod_{i=1}^{n} p^{(k)}(\mathbf{z}_i) = p^{(k)}(\mathbf{z}_c^*)\prod_{i=1}^{n} p^{(k)}(\mathbf{z}_i^*)|\det J_h|. \tag{11}$$

$$p^{(l)}(\mathbf{z}_c)\prod_{i=1}^{n} p^{(l)}(\mathbf{z}_i) = p^{(l)}(\mathbf{z}_c^*)\prod_{i=1}^{n} p^{(l)}(\mathbf{z}_i^*)|\det J_h|. \tag{12}$$

Taking quotients of Eq. (11) and (12), we have

$$\frac{p^{(k)}(\mathbf{z}_i)}{p^{(l)}(\mathbf{z}_i)} = \frac{p^{(k)}(\mathbf{z}_i^*)}{p^{(l)}(\mathbf{z}_i^*)}.$$

Here, we used the definition that $p(\mathbf{z}_c)$ and $p(\mathbf{z}_c^*)$ are invariant across different attribute labels, as well as the assumption that $\mathbf{A}^k$ and $\mathbf{A}^l$ differ only in the $i$-th entry. Now suppose $j \neq i$. Taking first-order derivative w.r.t $\mathbf{z}_j$ in the equation above yields

$$0 = \frac{\partial}{\partial \mathbf{z}_j}\left(\frac{p^{(k)}(\mathbf{z}_i^*)}{p^{(l)}(\mathbf{z}_i^*)}\right)$$

$$= \frac{\partial}{\partial \mathbf{z}_i^*}\left(\frac{p^{(k)}(\mathbf{z}_i^*)}{p^{(l)}(\mathbf{z}_i^*)}\right)\frac{\partial \mathbf{z}_i^*}{\partial \mathbf{z}_j}$$

$$= \frac{p^{(l)}(\mathbf{z}_i^*)\frac{\partial p^{(k)}(\mathbf{z}_i^*)}{\partial \mathbf{z}_i^*} - p^{(k)}(\mathbf{z}_i^*)\frac{\partial p^{(l)}(\mathbf{z}_i^*)}{\partial \mathbf{z}_i^*}}{p^{(l)}(\mathbf{z}_i^*)^2}\frac{\partial \mathbf{z}_i^*}{\partial \mathbf{z}_j}.$$

By Assumption A1, $p^{(l)}(\mathbf{z}_i^*)^2 \neq 0$. Therefore, we have

$$0 = \left( p^{(l)}(\mathbf{z}_i^*) \frac{\partial p^{(k)}(\mathbf{z}_i^*)}{\partial \mathbf{z}_i^*} - p^{(k)}(\mathbf{z}_i^*) \frac{\partial p^{(l)}(\mathbf{z}_i^*)}{\partial \mathbf{z}_i^*} \right) \frac{\partial \mathbf{z}_i^*}{\partial \mathbf{z}_j}$$

$$= \left( p(f_{i,0}(\epsilon_i^*)) \frac{\partial p(f_{i,1}(\epsilon_i^*))}{\partial \mathbf{z}_i^*} - p(f_{i,1}(\epsilon_i^*)) \frac{\partial p(f_{i,0}(\epsilon_i^*))}{\partial \mathbf{z}_i^*} \right) \frac{\partial \mathbf{z}_i^*}{\partial \mathbf{z}_j}$$

$$= \left( p(\epsilon_i^*) \left| \frac{\partial f_{i,0}(\epsilon_i^*)}{\partial \epsilon_i^*} \right|^{-1} \frac{\partial p(f_{i,1}(\epsilon_i^*))}{\partial \mathbf{z}_i^*} - p(\epsilon_i^*) \left| \frac{\partial f_{i,1}(\epsilon_i^*)}{\partial \epsilon_i^*} \right|^{-1} \frac{\partial p(f_{i,0}(\epsilon_i^*))}{\partial \mathbf{z}_i^*} \right) \frac{\partial \mathbf{z}_i^*}{\partial \mathbf{z}_j}.$$

Multiply both sides by $\left| \frac{\partial f_{i,1}(\epsilon_i^*)}{\partial \epsilon_i^*} \right| \left| \frac{\partial f_{i,0}(\epsilon_i^*)}{\partial \epsilon_i^*} \right|$ yields

$$0 = \left( \frac{\partial p(f_{i,1}(\epsilon_i^*))}{\partial \mathbf{z}_i^*} \left| \frac{\partial f_{i,1}(\epsilon_i^*)}{\partial \epsilon_i^*} \right| - \frac{\partial p(f_{i,0}(\epsilon_i^*))}{\partial \mathbf{z}_i^*} \left| \frac{\partial f_{i,0}(\epsilon_i^*)}{\partial \epsilon_i^*} \right| \right) p(\epsilon_i^*) \frac{\partial \mathbf{z}_i^*}{\partial \mathbf{z}_j}$$

$$= \left( \frac{\partial p(f_{i,1}(\epsilon_i^*))}{\partial \epsilon_i^*} \operatorname{sgn}\left( \frac{\partial f_{i,1}(\epsilon_i^*)}{\partial \epsilon_i^*} \right) - \frac{\partial p(f_{i,0}(\epsilon_i^*))}{\partial \epsilon_i^*} \operatorname{sgn}\left( \frac{\partial f_{i,0}(\epsilon_i^*)}{\partial \epsilon_i^*} \right) \right) p(\epsilon_i^*) \frac{\partial \mathbf{z}_i^*}{\partial \mathbf{z}_j}$$

$$= \left( \frac{\partial}{\partial \epsilon_i^*} \left( p(\epsilon_i^*) \left| \frac{\partial f_{i,1}(\epsilon_i^*)}{\partial \epsilon_i^*} \right|^{-1} \right) \operatorname{sgn}\left( \frac{\partial f_{i,1}(\epsilon_i^*)}{\partial \epsilon_i^*} \right) \right.$$

$$\left. - \frac{\partial}{\partial \epsilon_i^*} \left( p(\epsilon_i^*) \left| \frac{\partial f_{i,0}(\epsilon_i^*)}{\partial \epsilon_i^*} \right|^{-1} \right) \operatorname{sgn}\left( \frac{\partial f_{i,0}(\epsilon_i^*)}{\partial \epsilon_i^*} \right) \right) p(\epsilon_i^*) \frac{\partial \mathbf{z}_i^*}{\partial \mathbf{z}_j}$$

$$= \left( \frac{\partial}{\partial \epsilon_i^*} \left( p(\epsilon_i^*) \left( \frac{\partial f_{i,1}(\epsilon_i^*)}{\partial \epsilon_i^*} \right)^{-1} \right) - \frac{\partial}{\partial \epsilon_i^*} \left( p(\epsilon_i^*) \left( \frac{\partial f_{i,0}(\epsilon_i^*)}{\partial \epsilon_i^*} \right)^{-1} \right) \right) p(\epsilon_i^*) \frac{\partial \mathbf{z}_i^*}{\partial \mathbf{z}_j}$$

$$= \left( \left( \frac{p_{\epsilon_i^*}}{f'_{i,1}} \right)' - \left( \frac{p_{\epsilon_i^*}}{f'_{i,0}} \right)' \right) p_{\epsilon_i^*} \frac{\partial \mathbf{z}_i^*}{\partial \mathbf{z}_j},$$

where $\operatorname{sgn}(\cdot)$ is the sign function. By Assumptions A1 and A3, we have

$$\left( \left( \frac{p_{\epsilon_i^*}}{f'_{i,1}} \right)' - \left( \frac{p_{\epsilon_i^*}}{f'_{i,0}} \right)' \right) p_{\epsilon_i^*} \neq 0,$$

which implies

$$\frac{\partial \mathbf{z}_i^*}{\partial \mathbf{z}_j} = 0.$$

Since we are able to perform the above procedure for each $\mathbf{z}_i^*$ and each $\mathbf{z}_l$ where $l \neq i$, each $\mathbf{z}_i^*, i = 1, \ldots, n$ is solely a function of $\mathbf{z}_i$, i.e., $\mathbf{z}_i^* = h_i(\mathbf{z}_i)$. This implies that, the $i$-th row of the Jacobian matrix $\frac{\partial \mathbf{Z}^*}{\partial \mathbf{Z}}$ has only one nonzero entry for $i = 1, \ldots, n$. Denote by $\mathbf{z}_{[n]} = (\mathbf{z}_1, \ldots, \mathbf{z}_n)$ and $\mathbf{z}_{[n]}^* = (\mathbf{z}_1^*, \ldots, \mathbf{z}_n^*)$. The above derivation indicates $\frac{\partial \mathbf{z}_{[n]}^*}{\partial \mathbf{z}_c} = 0$. Since $h$ is a diffeomorphism and $\frac{\partial \mathbf{Z}^*}{\partial \mathbf{Z}}$ is of full rank, the matrix $\frac{\partial \mathbf{z}_{[n]}^*}{\partial \mathbf{z}_{[n]}}$ must also be of full row rank because $\frac{\partial \mathbf{z}_{[n]}^*}{\partial \mathbf{z}_c} = 0$. This indicates that $h_i$ is invertible, i.e., $\mathbf{z}_i = h_i^{-1}(\mathbf{z}_i^*)$. Therefore, $\mathbf{z}_i$ is solely a function of $\mathbf{z}_i^*$. $\qquad \square$

# D    CAUSAL CONDITIONING TO HANDLE DEPENDENT LABELS

In the main paper, we present the causal conditioning method to handle the causally related labels. Specifically, we perform causal discovery. Then for each node, we use its parents as conditioning to construct new labels. Then we use the new label to train the generative. During inference, we can assign the value to the column according to the parents' value. For instance, given age $\rightarrow$ eyeglasses, changing the eyeglasses still lead to age-related changes. After the label transformation, we are now provided with three columns (Age, Eyeglass—Age=old, Eyeglass—Age=1). If a user assigns age=old and wearing eyeglasses, then our label would be [1,1,0] since we conditional on the age information.

Causal conditioning separates the information of the child node from the shared information with its parents. Here we provide examples to show that such causal conditioning is effective in Figure 11.

Without Causal          With Causal

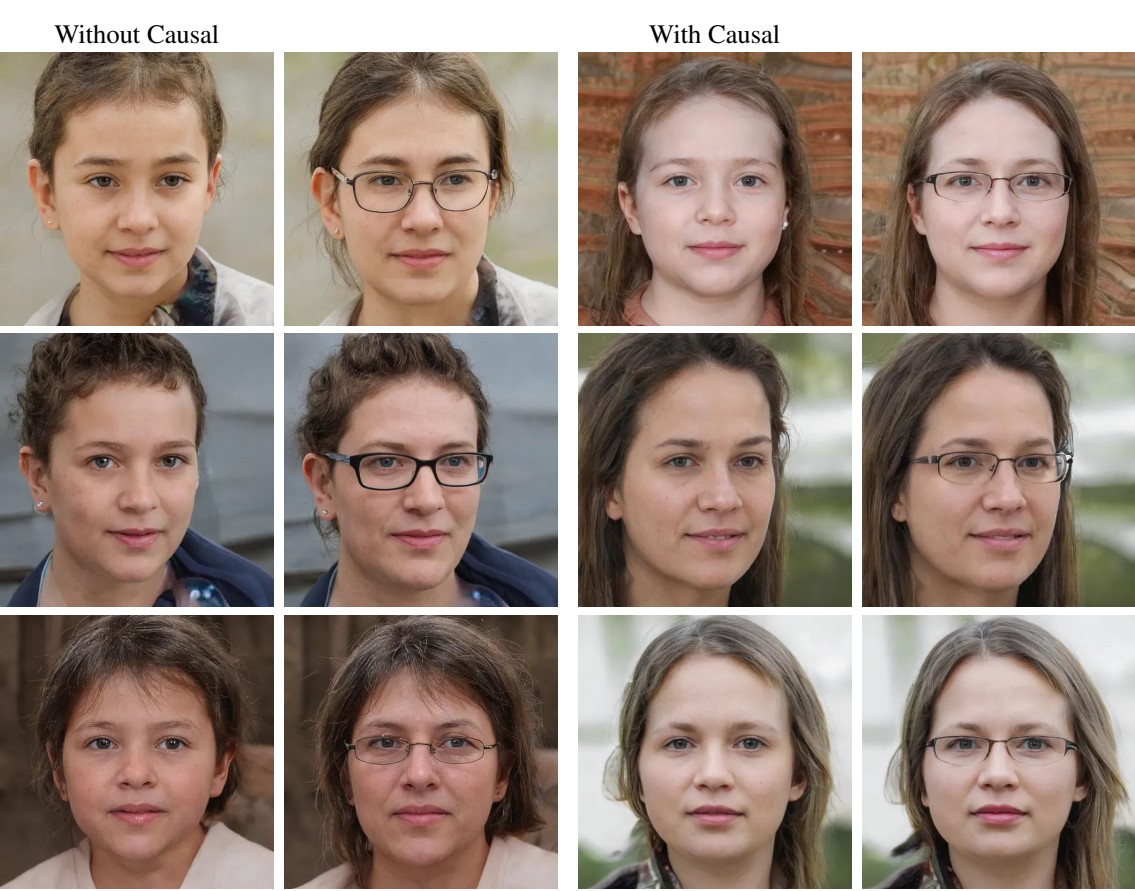

Figure 11: The effects of causal conditioning. Left two columns: no causal conditioning. Right two columns: with causal conditoning. Through causal conditioning, our model learns to factorize the informatiohn about eyeglasses only.

StyleADA (Karras et al., 2020a)   AugGAN (Hou et al., 2024)   Ours

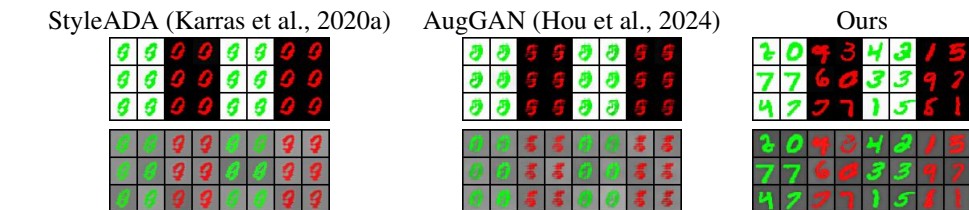

Figure 12: MNIST generation results. Given limited training images, the baselines suffer the mode collapse problem. By activating the white background and black background together, we can obtain gray MNIST images.

## E    MORE RESULTS IN GENERATION

In this section, we present more generation results.

## F    MORE EDITING RESULTS

We present more real-world image editing cases in Figure 15.

## G    IMPLEMENTATION DETAILS

We provide the training code in the supplementary material. We build our method on StyleADA (Karras et al., 2020a). Our main empirical contribution lies in the redesigning of the mapping network used in StyleADA. In StyleGAN-based methods, the Gaussian noises and a class embedding are fed together into a MLP to get an entangled space $\mathcal{W}$. In this paper, we build our model based on the minimal change principle. For each attribute $A_i$, we use a 2-layer MLP to transform the input noise $\epsilon_i$ into activated or deactivated concept. Then we concatenate the outputs together to get a latent **z**. In other words, we have a $\mathcal{Z}$ space instead of a $\mathcal{W}$ space now.

For each attribute, we set the dimension to be 20 at the beginning of training. We also employ a learnable mask $m_i$ to encourage the model select the dimension for each concept. We apply $L_1$ sparsity on the mask. We usually set $\lambda_{sparsity} = 0.1$.

As for the real image editing task, we choose to extend the $\mathcal{Z}$ space into $\mathcal{Z}^+$ space following the previous StyleGAN inversion methods Abdal et al. (2019; 2021). Unlike $\mathcal{Z} \in \mathbb{R}^{1,d}$, now the $\mathcal{Z}^+$ space has $\mathbb{R}^{k,d}$ where $k$ is the number of repeated for the latent variables. For example, in training FFHQ dataset, StyleGAN based approaches repeat the latents 16 times. So, now we estimated noise variables $\{\hat{\epsilon}\}$ has shape $16 \times d$. Then we optimize the variables using loss function mentioned in the main paper. As for $\mathcal{L}_{reg}$, we also regularize the noise maps to penalize them from carrying incoherent signals following StyleADA (Karras et al., 2020b). For more details of the regularization, we refer readers Section 5 in (Karras et al., 2020b).

StyleADA(Karras et al., 2020a)    AugGAN (Hou et al., 2024)    Ours

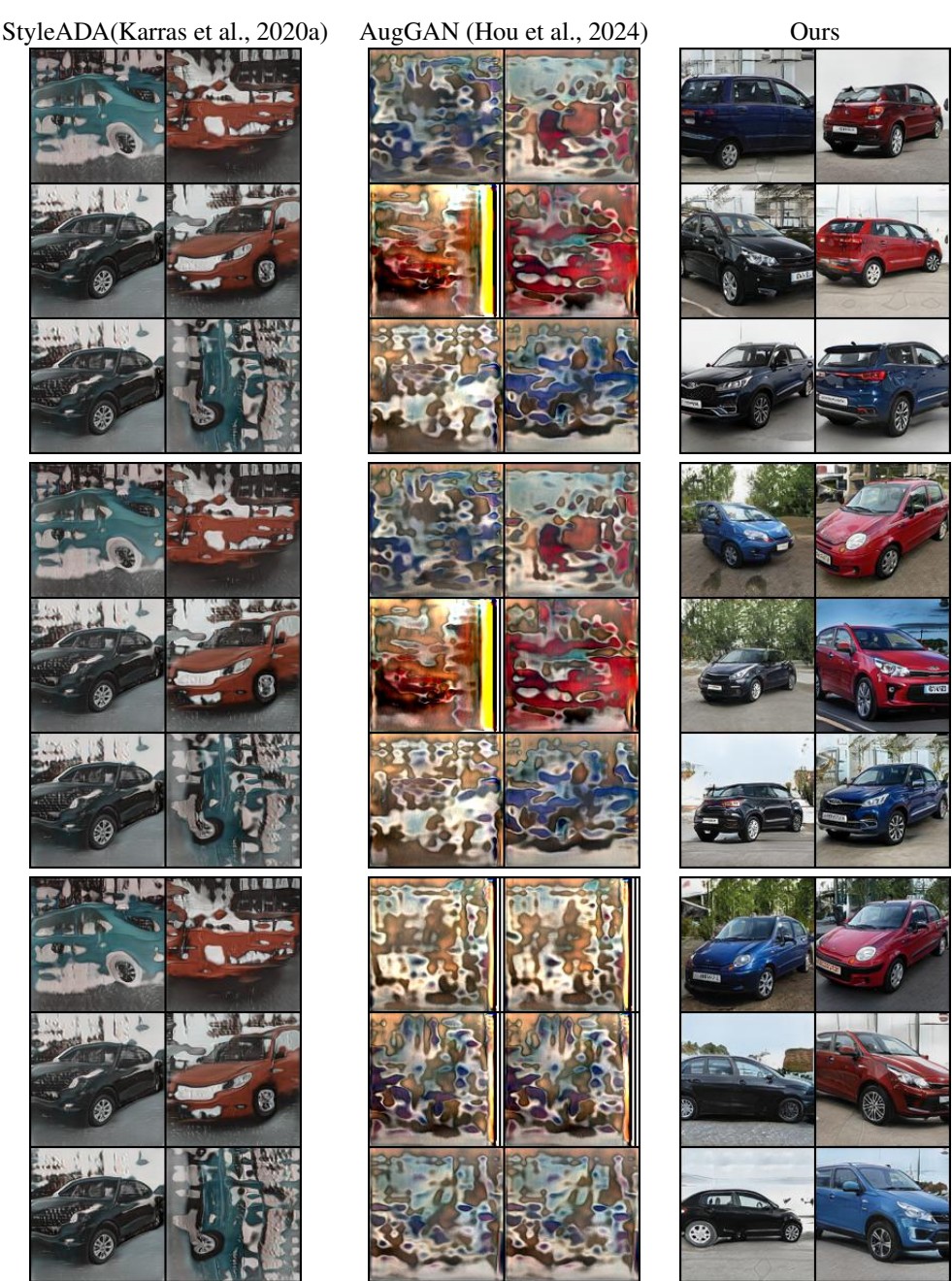

Figure 13: The in-domain image generation. We need to generate blue, red matiz, black, red Rio, and black blue Tiggo. Unfortunately, the baseline models suffer the mode collapse and fail to generate natural images.

StyleADA(Karras et al., 2020a)    AugGAN (Hou et al., 2024)    Ours

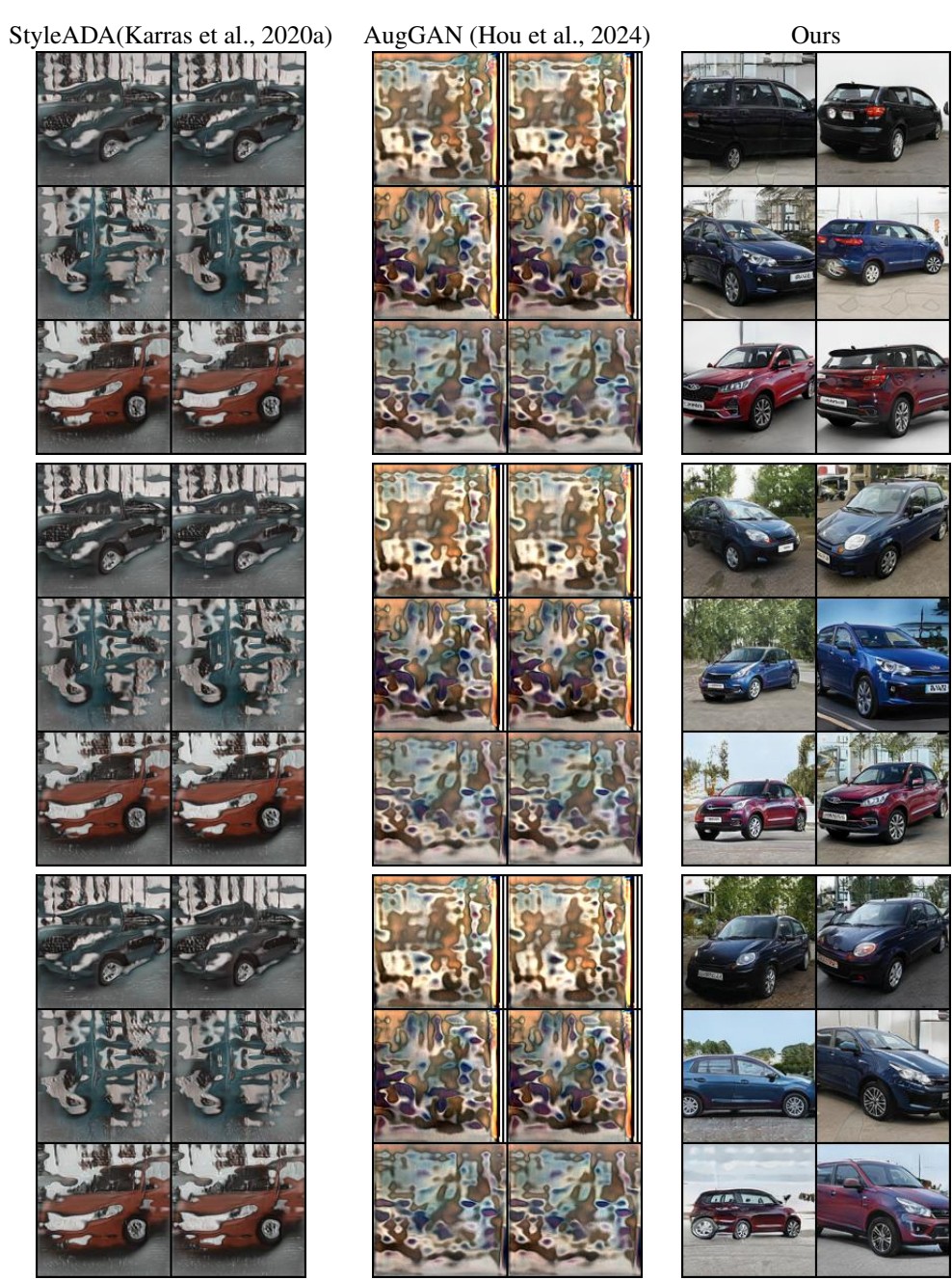

Figure 14: The out-of-domain image generation. We need to generate black matiz, blue Rio, and red Tiggo. Unfortunately, the baseline models suffer the mode collapse and fail to generate natural images.

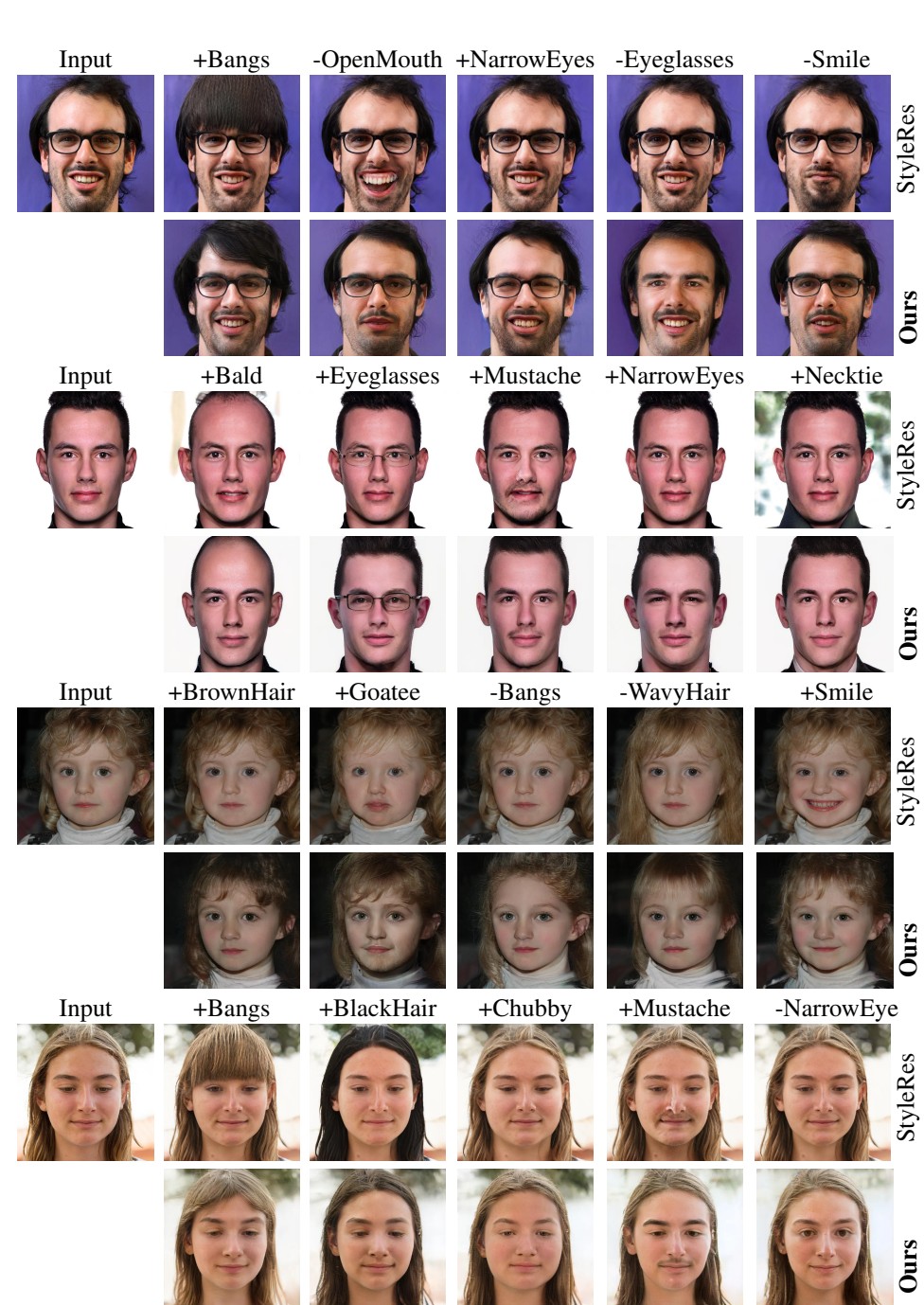

Figure 15: Comparisons on real image editing task. The strong baseline StyleRes(Pehlivan et al., 2023) achieves good reconstruction in details with a higher-dimensional feature while failing to follow the conditions when the given conditions are uncommon, e.g., wearing necktie and opening the eyes.

