# OpenReview forum: "Learning Identifiable Concepts for Compositional Image Generation"
_ICLR.cc/2025/Conference — ICLR 2025 Conference Withdrawn Submission_

### Official Review · Reviewer_3gsM · 2024-10-23

**Soundness:** 2
**Presentation:** 1
**Contribution:** 2
**Rating:** 3
**Confidence:** 4

**Summary:**

The authors study compositional attribute generation and editing in image synthesis models. They argue relatively convincingly that the current large-scale image generation models fail to generate uncommon attribute labels (e.g. “female” + “facial hair”), and propose a methodology to address this through the use of masks learned with a causal structure. The results show the method produces images that do not exhibit mode collapse like the baselines. In the case of editing real images, there is significant improvements to the editing of rare attribute combinations over recent work.

**Strengths:**

- The studied problem of generating unseen attribute combinations is a pertinent one, with important implications for under-represented demographics and subpopulations. The authors did a convincing job with Figure 1 and in the introduction of motivating the problem with current large-scale image synthesis models, and making the benefits of the proposed solution salient.
- I appreciate that the experiments are relatively thorough in exploring multiple forms of image synthesis. Not only do the authors consider unconditional synthesis, but they also show how one can edit real images, greatly improving the contribution of their method.

**Weaknesses:**

## [W1] Trade-off in performance for common attributes

In Figure 7, whilst the method clearly excels at generating rare attribute combinations (e.g. female + goatee), it fails in other cases to make more common edits (e.g. +blonde hair, or +bald).

To me, this seems very problematic. Almost by definition, most users will be interested in generating common attribute combinations. The fact that the method works so well on unseen combinations is a testament to its potential value, to be clear, but trading-off functionality for common edits at the same time seems like a clear and fundamental limitation. What use case does the proposed method serve if it’s at the cost of the common attribute combinations? In my view, this is the primary issue with the paper.

At minimum, I would expect to see a detailed discussion of this trade-off, and a solid justification for why it is worth making. Do the authors have any insights into why this might be happening? Furthermore, an insightful study would be one that quantifies the "accuracy" of edits for common vs uncommon attributes -- one could train a CelebA binary classifier to classify if an edited image actually depicts the new attribute or not, and one could see a breakdown of the performance for common vs rare attributes.

## [W2] Lack of convincing baselines for independent attribute datasets

I am not convinced that the authors do a good job of showcasing the benefits of their method in the independent attribute setting (Table 1 and Figure 5). Concretely, it is worrying that the baseline methods mostly fail to generate anything coherent at all (~20x as large FID scores). This really does not tell us much other than the baselines failed to train well (which could be for any number of reasons).

The authors could do a better job training the baseline models for a fairer comparison (e.g. perhaps with significant data augmentation, or through differentiable techniques such as [1]). Ultimately, we are not interested in the image quality itself, but instead in how well they perform in the “Out-FID” row on the rare attribute combinations. Through better training of the base models, we can isolate the impact of the proposed method on this row of interest without the confounding variable of the raw image synthesis quality in the way.

## minor

The paper is full of typos, and some poorly written sentences. Just to mention a handful of examples from the introduction alone on the second page:

- [L64] leads to → lead to
- [L66] Ssadow → Shadow
- [L72] mkae→ make

Ultimately these typos are indicative of a lack of care for presentation, and at times this renders the sentences hard to parse which I found often detracting from the content of the paper. I suggest some careful proof-reading is needed before the camera-ready or resubmission.

---

[1] Zhao et al. “Differentiable Augmentation for Data-Efficient GAN Training.” NeurIPS 2020.

**Questions:**

It seems a relatively big limitation that the method relies on such rigid one-hot labels when the modern paradigm of image editing involves free-form textual descriptions. Do the authors envision easy ways to extend this to continuous or multi-label attributes, or free-form text? A discussion of the proposed binary attribute paradigm relates to the common free-form text editing one -- and their relative strengths -- would be insightful here.

---

### Official Review · Reviewer_X3TZ · 2024-10-31

**Soundness:** 3
**Presentation:** 2
**Contribution:** 2
**Rating:** 5
**Confidence:** 3

**Summary:**

This paper proposes a GAN-based framework for learning identifiable concepts. Given ground-truth attribute labels, random noise is transformed into latent representations aligned with these labels, and sparsified using learnable masks to enforce a minimal change principle. To mitigate existing correlations between certain attributes, the authors explicitly identify causal relationships among attributes and factorize the labels to remove dependencies. Empirical results demonstrate that the proposed method outperforms baselines in terms of data efficiency and controllability.
The main contributions of the paper are as follows:
- Formulation of the minimal change principle to learn compositional concepts, along with an efficient approach to factorize causally related attributes.
- Theoretical proof that the proposed method can recover ground-truth concepts.
- Empirical evidence showcasing improved data efficiency and controllability.

**Strengths:**

- The idea of transforming labels to identify and disentangle causal relationships among attributes is interesting, and the authors have effectively demonstrated its impact in the experimental results.
- The proposed method significantly outperforms the baselines, both qualitatively and quantitatively, validating its practical advantages in achieving high-quality, controllable image generation, even in low-data settings.

**Weaknesses:**

- It is unclear how the proposed method learns compositional concepts more effectively or in a fundamentally different way compared to existing approaches. Since the baselines also leverage disentangled ground-truth attribute labels, wouldn’t they similarly be capable of learning a generative model for compositional generation? In a similar context, it’s not fully explained why the proposed method is more data-efficient than the baselines. A more detailed elaboration on these points would strengthen the paper.
- The paper introduces several components (e.g., sparsity loss, learnable masks, $\mathbf{z}^{\text{null}}_i$​), but the justification for each component and their connections seems weak. It is a bit confusing as a reader to understand why each part is necessary. Please refer to the questions below for specific points on this aspect.

**Questions:**

- What is the role of $\mathbf{z}_c^*$ in equation (1)?  It seems like it should encode information not represented by annotated labels (e.g., nuanced details). However, isn’t this type of information typically handled by the random noise $\epsilon$? Does including $\mathbf{z}_c^*$ have a significant impact on performance?
- What is the role of $\mathbf{z}^{\text{null}}_i$ in equation (6)? What kind of information is it intended to encode?
- It is hard to fully understand why enforcing the sparsity loss in equation (7) induces the minimal change principle. While Lines 522–524 suggest that constraining the representation’s dimensionality limits redundant information, this rationale is not entirely convincing. The minimal change principle, as described by the authors, states that "the influence brought by each ground-truth concept should be minimal," which implies that changes in representation space should translate to minimal changes in the output space (e.g., altering the ‘age’ should yield the same image but with a different age). However, the sparsity loss in Equation (7) seems to restrict the input representation space rather than the changes in the output space, making it unclear how this connects to the minimal change principle.
- It would be better to use distinct notations for $\mathbf{z}_i$ in equation (3) and (6) as they are clearly denoting different variables.
- Does $\mathbf{m}_i$ in L177 refer to $\mathbf{A}_i$?
- In Figure 6, the authors claim that foundation models (e.g., GPT-4o) generate unrealistic images for unseen attribute combinations. However, all images generated by GPT-4o in Figure 6 appear unnatural, suggesting that the poor results might not be due to rare attribute combinations but other factors, such as improper prompts provided to the model. Could the authors clarify if proper prompt was used, and whether different prompts might correct GPT-4’s performance on unseen combinations?
- In Table 8, which evaluates generation performance on human faces, it would be more comprehensive to include metrics for other generative models (e.g., GPT-4o, Meta AI, Stable Diffusion 3, as in Figure 6) for comparison.
- Between the sparsity condition and causal conditioning, which component is the key factor that causes the proposed method to succeed where the baselines fail in Figure 5? Would simply applying causal conditioning to the baselines improve their performance?

---

### Official Review · Reviewer_A5LF · 2024-11-02

**Soundness:** 2
**Presentation:** 2
**Contribution:** 2
**Rating:** 3
**Confidence:** 3

**Summary:**

This paper presents the minimal change principle and causal conditioning to allow generative models to create compositional images with clear, identifiable concepts. The central idea is to control image attributes without inducing unintended changes. To accomplish this, the authors regularize the model to learn the minimum dimensions needed to edit an attribute and use causal discovery algorithms to disentangle dependent attributes. The authors empirically and theoretically demonstrate that this approach enables models to learn attributes that are both identifiable and composable.

**Strengths:**

* The minimal change principle is intuitive and makes sense.
* The concept of causal conditioning is interesting and intuitive.
* The proposed method achieves superior FID scores on MNIST4 and Car9 datasets compared to StyleADA and AugGAN.

**Weaknesses:**

* There is no quantitative comparison showing if the model controls attributes better than the baselines. For example, metrics like Editing
 FID from StyleRes could be used to demonstrate controllability.
* Baselines about image editing and compositional image generation are missing.
  * CausalGAN (Kocaoglu, et al. "Causalgan: Learning causal implicit generative models with adversarial training." 2017.)
  * AugGAN(on FFHQ) (Hou, et al. "Augmentation-aware self-supervision for data-efficient GAN training." 2024.)
  * StyleRes (Pehlivan, et al. "Styleres: Transforming the residuals for real image editing with stylegan." 2023.)
  * HyperStyle (Alaluf, et al. "Hyperstyle: Stylegan inversion with hypernetworks for real image editing." 2022.)
  * StyleTransformer. (Hu, et al. "Style transformer for image inversion and editing." 2022.)
* Except for Figure 8, there is no metric provided for editability or composability, making it difficult to assess whether the proposed method learns more identifiable concepts than the baselines. Additionally, in the ablation studies, it is challenging to gauge the effectiveness of the proposed components without metrics for editability or composability.

**Questions:**

* Regarding Section 3.4, it’s unclear why inversion cannot be done in the $z$ space or $w$ space. Would it be possible to move the input of $f\_i$ to $z$ or $w$ space and perform inversion in $z$ instead?
* It is unclear why the first row of Table 2 is labeled as "Ours." It appears to correspond to StyleGAN2-ADA.

---

### Official Review · Reviewer_Apr6 · 2024-11-03

**Soundness:** 3
**Presentation:** 3
**Contribution:** 3
**Rating:** 6
**Confidence:** 4

**Summary:**

This paper addresses an intriguing problem: compositional image generation. It introduces the minimal change principle and proposes a method to limit the information introduced by each label. A causal conditioning approach is employed to disentangle concepts from correlations. The effectiveness of this method is validated across several tasks.

**Strengths:**

- Compositional image generation is a critical and practical problem, and this paper proposes a method to address it.

- The paper presents an identifiable guarantee for learning the underlying concepts.

- The generated images are promising, demonstrating the potential of the proposed method.

**Weaknesses:**

1. This method relies on pre-defined attributes, which limits the method's practical applicability.

2. Additionally, the proposed methods are evaluated only on simple datasets, which may not adequately represent complex real-world scenarios.

**Questions:**

1. When the attributes of a dataset are not directly accessible, how can they be retrieved?

2. The current method utilizes a GAN-based model as the foundation. Is it feasible to implement this approach using a diffusion model instead?

3. Additionally, how many attributes can this method manage effectively? If we aim to train a general-purpose model that can handle more than a thousand attributes, what strategies should be employed to address this scenario?

---

### Note · Authors · 2024-11-15

**Comment:**

We thank every reviewer for your thoughtful feedbacks and we will edit our manuscript according to your suggestions. Thanks!

**Withdrawal Confirmation:**

I have read and agree with the venue's withdrawal policy on behalf of myself and my co-authors.